# Functionalization of Photosensitized Silica Nanoparticles for Advanced Photodynamic Therapy of Cancer

**DOI:** 10.3390/ijms22126618

**Published:** 2021-06-21

**Authors:** Ruth Prieto-Montero, Alejandro Prieto-Castañeda, Alberto Katsumiti, Miren P. Cajaraville, Antonia R. Agarrabeitia, María J. Ortiz, Virginia Martínez-Martínez

**Affiliations:** 1Departamento de Química Física, Universidad del País Vasco/Euskal Herriko Unibertsitatea (UPV/EHU), 48080 Bilbao, Spain; ruth.prieto@ehu.eus; 2Departamento de Química Orgánica, Facultad de CC. Químicas, Universidad Complutense de Madrid, 28040 Madrid, Spain; alprieto@ucm.es (A.P.-C.); agarrabe@quim.ucm.es (A.R.A.); 3CBET Research Group, Department Zoology and Animal Cell Biology, Faculty of Science and Technology and Research Centre for Experimental Marine Biology and Biotechnology PiE, University of the Basque Country UPV/EHU, 48620 Basque Country, Spain; mirenp.cajaraville@ehu.eus (M.P.C.); 4GAIKER Technology Centre, Basque Research and Technology Alliance (BRTA), 48170 Zamudio, Spain; katsumiti@gaiker.es (A.K.)

**Keywords:** BODIPY-based photosensitizers, functionalized silica nanoparticles, folic acid, PEG, photodynamic therapy, HeLa cells, (photo) toxicity

## Abstract

BODIPY dyes have recently attracted attention as potential photosensitizers. In this work, commercial and novel photosensitizers (PSs) based on BODIPY chromophores (haloBODIPYs and orthogonal dimers strategically designed with intense bands in the blue, green or red region of the visible spectra and high singlet oxygen production) were covalently linked to mesoporous silica nanoparticles (MSNs) further functionalized with PEG and folic acid (FA). MSNs approximately 50 nm in size with different functional groups were synthesized to allow multiple alternatives of PS-PEG-FA decoration of their external surface. Different combinations varying the type of PS (commercial Rose Bengal, Thionine and Chlorine e6 or custom-made BODIPY-based), the linkage design, and the length of PEG are detailed. All the nanosystems were physicochemically characterized (morphology, diameter, size distribution and PS loaded amount) and photophysically studied (absorption capacity, fluorescence efficiency, and singlet oxygen production) in suspension. For the most promising PS-PEG-FA silica nanoplatforms, the biocompatibility in dark conditions and the phototoxicity under suitable irradiation wavelengths (blue, green, or red) at regulated light doses (10–15 J/cm^2^) were compared with PSs free in solution in HeLa cells in vitro.

## 1. Introduction

Currently, several alternatives are used to treat cancer, including surgery and chemo, radio- or immune-therapy, although depending on the type of cancer an effective method has not been found yet. In this regard, Photodynamic Therapy (PDT) is a complementary treatment that can be applied as a combined therapy to enhance anticancer efficiency by a synergic or additive effect with conventional methods. PDT involves a light source, a photosensitizer (PS), and oxygen. During PDT, PS is activated under light at a specific wavelength to generate reactive oxygen species (ROS), mainly singlet oxygen (^1^O_2_), a cytotoxic species able to promote apoptosis or necrosis of cancer cells [1]. Nowadays, preclinical and clinical trials have proven PDT to be effective in early-stage tumors or the palliation of advanced cancers, such as skin, head, neck, esophageal, or lung cancer, improving patient survival [2,3,4,5]. PDT is considered a less invasive and more precise treatment (locally controlled by the light irradiation of malignant tissue), without inducing long-term side effects, and it has a lower cost with respect to other treatments. Nevertheless, the limitations of PDT are mainly related to the availability of the PSs. Despite there being several PSs approved by the FDA, most of them are hydrophobic and/or tend to have poor selectivity to malignant tissues [6,7,8,9].

The ideal PS to be used as a photoactive drug for PDT should be non-cytotoxic in dark conditions, selective to cancer tissues, and display limited stability in vivo to minimize side effects; it should have intense absorption bands (*ε* ≥ 50,000 M^−1^ cm^−1^), preferentially in the phototherapeutic window to ensure deeper penetration of light into tissues [10] (630–850 nm), and high singlet oxygen production to reduce doses and irradiation time; it should be photoresistant to avoid the photodecomposition of the PS during treatment; and finally, it should present an amphiphilic nature, being soluble in water as well as permeable through the cell membrane. At the moment, few PSs fulfill these requirements, and new molecular designs are required [6,7,11,12,13,14]. One approach for obtaining new molecules is focused on the synthesis of improved PS to overcome their limitations, but this usually requires multistep chemistry, increasing the costs and production time, hampering the implementation for clinical uses. In this context, BODIPY dyes have recently attracted attention as potential photosensitizers [15,16,17,18]. They are characterized by intense absorption and emission bands in the green region, and resistance to photobleaching [19,20]. Despite being highly fluorescent chromophores (antagonistic property to ROS generation) and poorly soluble in water, their synthesis allows easy, versatile, and selective modification of their molecular structure to increase the population of the triplet state, and consequently their singlet oxygen generation, while also shifting their spectroscopic bands into the clinical window. These modifications include the addition of iodine heavy atoms, *π*-conjugated systems in the BODIPY skeleton, or the design of orthogonal BODIPY dimers [15,17,18,21,22,23,24,25,26,27,28,29,30,31,32,33]. Further functionalization of the BODIPY chromophore is related with the incorporations of different targets to increase their solubility in aqueous media and enhance their selectivity to cancer cells [14,34,35,36,37,38,39].

Another alternative is the use of nanomaterials as (photo)drug carriers. They have a large surface-to-volume ratio, which allows the administration of a large amount of active components, preventing their degradation or inactivation by plasma components, delivering soluble and stable formulations in aqueous media, and enhancing their accumulation inside tumor tissues by so-called passive targeting due to the enhanced permeability and retention (EPR) effect [11,40,41,42,43,44,45,46,47,48,49]. Additionally, the selectivity to cancer cells can be improved by active targeting through surface modifications with target ligands, such as proteins, polysaccharides, nucleic acids, peptides and small molecules that bind to specific receptors overexpressed on the surface of malignant cells but not on healthy cells [9,50,51,52].

Currently, there are many different types of nanoparticles based on liposomes, polymeric, micellar, metallic, or protein for medical use [40,41,42,43,44,45,46,53,54,55]. In this regard, silica nanoparticles (SN) have attracted attention as carriers for drug delivery due to their properties, which include reduced toxicity, good biocompatibility, high surface area, easy functionalization, optical transparency, and low cost [56,57]. PS-loaded silica nanoparticles have been reported as promising singlet oxygen generator platforms, improving the photoactive drug delivery by enhancing PS poor solubility and selectivity for cancer cells [58,59,60,61,62]. The PS can be physically encapsulated or covalently attached to the internal or external surface of the silica nanoparticles [63,64,65,66,67]. Briefly, loading PS within the nanostructure ensures a high photostability but restrains the diffusion of oxygen species (molecular oxygen towards inside and singlet oxygen towards outside). It has been demonstrated that nanoparticles with draped-PS outside lead to better ^1^O_2_ productivity than PS located inside [68,69,70].

In the last few years, diverse nanoplatform designs have been used as vehicles to carry BODIPY-PS [71,72,73,74,75,76,77,78,79,80,81], or even BODIPY-based nanoparticles, through the self-assembly process [79,80,82,83,84,85,86,87,88]. However, despite the advantageous properties of SN, mentioned above, few examples can be found in the literature of their use as carriers for BODIPY-PSs [89,90].

In this work, different PSs (Figure 1) were tethered to the external surface of 50 nm MSNs. First, three commercially available PSs, Rose Bengal (RB), Chlorin e6 (C6), and Thionine (Th), recognized as suitable singlet oxygen generators and extensively employed in PDT, were used [6,9,10,11,59,91,92,93]. These dyes already have functional groups in their molecular structure (carboxylic in Rose Bengal and Chlorin e6 and amine in Thionine) and can be easily grafted to the external surface of MSNs. Afterward, seven custom-made BODIPY-based PSs were used, which were rationally designed to effectively generate high singlet oxygen production under illumination at different wavelengths of the visible spectra (blue, green or red light) [12,18,24,31,94,95], and to endow suitable graftable groups to be anchored at MSNs.

Additionally, MSNs were externally coated with polyethylene glycol (PEG), as it is usually required to stabilize nanoparticle systems, enhance their life-time in the blood system and avoid the induction of immune responses [44,47,49,94,95,96,97,98]. For that, several PEG derivatives (Figure 1) with different graftable groups at one end of the chain (succinimide group or with silyl group Si-PEG) were tethered at the MSN shell. The length of the polymer chain (750 Da, 2000 Da, and 5000 Da) was also adjusted to improve nanoparticle stabilization in water.

Finally, the selectivity of MSNs for cancer cells was enhanced by the addition of a peripheral target for cancer cells. Folic acid (FA), a low-cost, stable and small molecule with available functional groups (Figure 1), is widely used to target several types of cancer cells, in particular, those overexpressing folate receptors (FR) on their surface, such as ovarian, endometrial and kidney cancer cells [99,100]. Thus, tethering FA on the nanoparticle’s surface promotes a higher cellular uptake via endocytosis [101,102,103,104]. In fact, our recently published study quantitatively demonstrated a significantly higher accumulation of FA-functionalized fluorescent MSNs compared to nanoparticles without FA in HeLa cells [105].

The linkage between the PSs and the MSNs and the length of PEG were firstly optimized with the commercial PS Rose Bengal and then implemented for the rest of PSs. The physicochemical features (morphology, diameter, size distribution, PS loaded amount) and photophysical properties (absorption capacity, fluorescence efficiency, and singlet oxygen production) were detailed. The efficiency of the PS-PEG-FA MSN nanoplatforms was tested in HeLa cells in vitro, and the results were compared with those obtained in cells exposed to PSs free in solution.

## 2. Results and Discussion

### 2.1. Silica Nanoparticles Characterization

Mesoporous silica nanoparticles with a suitable size for medical applications and particularly for PDT [98,106] were synthesized by the modified Stöber method [107] as described elsewhere [105]. The external surface of mesoporous nanoparticles surface was functionalized with amino group (NH-MSN) or carboxylic group (COOH-MSN). The latter type was obtained after conversion of CN-MSN in acidic conditions, according to the synthesis route described in Materials and methods section and Supporting Material. Bare MSNs analyzed by SEM and TEM showed spherical morphology and mesoporous structure (Figure 2 and Appendix A), with a size distribution of 50 ± 10 nm. The external functionalization of MSNs was studied by XPS (Appendix A). The presence of 5% of nitrogen atoms in both NH-MSN and CN-MSN confirmed the existence of amine/cyano functional groups located outside of the nanoparticles whereas the absence of nitrogen atoms in the COOH-MSN indicates an effective conversion of CN into COOH groups. In the case of FTIR spectra (Appendix A), an intense peak located at υ = 1110–1000 cm^−1^ as well as a wide band placed at υ = 3650–3200 cm^−1^ were recorded in every sample spectrum, and are assigned to Si-O-C and O-H groups, respectively. A characteristic band of cyane group (C≡N) at υ = 2260–2240 cm^−1^ was recorded in CN-MSN, which disappeared in the COOH-MSN system (Appendix A blue), indicating again the total conversion from -CN to -COOH. Furthermore, the typical band of COOH group (COO-H υ = 3550–2550 cm^−1^, C=O υ = 1775–1650 cm^−1^) was also registered in COOH-MSN.

The sizes of the three types of nanosystems were also characterized in water suspension by DLS. Both NH-MSN and COOH-MSN showed similar hydrodynamic diameter, of around 60–70 nm, to the size of the nanoparticles by TEM (Table 1), whereas the larger diameter, derived for CN-MSN, of 280 nm, indicates a tendency to form aggregates. Zeta potential values obtained for CN-MSN but also for NH-MSN (≤±25 mv) [108] confirm a poorer stability in water with respect to COOH-MSN system. The higher stability of this later functionalized COOH-MSN is attributed to the presence of carboxylic groups at the external surface, partially deprotonated (COO^−^) in aqueous media and the superficial negative charge makes the nanosystem more stable by electrostatic repulsion, Table 1.

Actually, the MSNs’ stability in aqueous media is certainly controlled by the type and the number of molecules lodged at the external surface. It has been demonstrated that the presence of organic PSs makes the external surface of MNS more hydrophobic, promoting nanoparticles agglomeration, hindering their stability in aqueous media [109]. Note here that the particle–particle aggregation is detrimental to singlet oxygen production and should be avoided or minimized. To optimize the PS loading and to ensure the stability of the nanoparticles in suspension, several syntheses were carried out for the amine-functionalized MSN (NH-MSN), employing commercial RB as PS and different PEG derivatives (RB-PEG_n_-NH-MSN). The combinations were focused on the variation of (i) the functional group of the MSN (OH-, or NH_2_-) at which PS and PEG molecules were attached, and (ii) the length of PEG chain (750 Da, 2000 Da, and 5000 Da).

### 2.2. Optimization of the Functionalization of Silica Nanoparticles with Rose Bengal as PS

Rose Bengal is a commercial PS with an intense absorption band (*λ*_max_ = 556 nm; *ε* = 9.8·10^4^ M^−1^cm^−1^) and high singlet oxygen production (Φ_∆_ = 0.86 in CH_3_OD). The carboxylic function in the RB molecular structure allows the covalent grafting to be inserted to amine groups or to the intrinsic hydroxyl groups of the external surface of MSN [109]. Nevertheless, both RB-MSN nanosystems (RB grafted at the external NH_2_ or OH) showed instantaneous flocculation in water media and the suspension was only viable in less polar solvents. Since stable nanoparticle suspension in water is crucial to obtain competitive hybrid nanocarriers for PDT [109], pegylation of the outside of MSNs is required to avoid the precipitation of the nanoparticles. Firstly, to optimize the stabilization of the system, NHS-PEG of different chain length (750 Da, 2000 Da and 5000 Da) were linked to the amine groups of the silica, while RB was anchored in OH groups (samples RB-PEG_750_-NP-**a**, RB-PEG_2000_-NP-**a**, and RB-PEG_5000_-NP-**a** in Table 2).

According to zeta potential (Table 2), the least favored value (−4.3 mV) was registered for sample RB-PEG750-NP-a with the shortest PEG chain in this series, indicating its inefficiency at improving the stability of RB-MSN in water. Indeed, a similar value of around -4 mV was obtained for NH-MSN without RB (Table 1). In contrast, PEG of higher molecular weight, 2000 Da and 5000 Da (RB-PEG_2000_-NP-**a** and RB-PEG_5000_-NP-**a** in Table 2) rendered Zpot values of −25 mV, indicating good stability of these nanosystems in water. The longer PEG-5000 did not lead to an improvement of the stability with respect to PEG-2000, which could likely be assigned to a different conformation adopted at the external surface [110]. Additionally, long PEG chains can also impede the internalization of nanoparticles into the cells [111,112]. Thus, a PEG of 2000 Da was selected as the most suitable, and was employed in the rest of the samples.

Next, different anchorages between PEG and MSN (at a fixed PEG length of 2000 Da) were also tested. The anchoring of Si-PEG (silylated PEG of 2000 Da, Figure 1) to the external OH-groups of MSN, samples RB-NP-**b** and RB-NP-**c**, led to even higher Zpot values with respect to sample RB-PEG_2000_-NP-**a** (with PEG at the amine groups), Table 2. This fact is possibly due to a higher presence of PEG at the surface because there are more accessible OH-groups than NH_2_-groups at the silica external surface [97]. This assumption was confirmed for RB, showing a double dye loading when was tethered to OH with respect to NH_2_ groups of the MSN external surface (RB-PEG-NP-b vs. RB-PEG-NP-**c** in Table 2) [109].

The stability of the nanoparticles can also be studied by the absorption spectra of the RB-PEG-MSNs samples in water suspension (Figure 3). The registered bands for RB-PEG_2000_-**a** and RB-PEG_5000_-**a**, practically identical, showed more prominent shoulders at both sides of the main absorption band, indicative of a higher dye aggregation tendency. Indeed, according to the absorption spectra, the dye aggregation follows the tendency RB-PEG_5000_-NP-**a** ≈ RB-PEG_2000_-NP-**a** > RB-PEG-NP-**c** > RB-PEG-NP-**b**. For the samples RB-PEG_2000_-NP-**a**, RB-PEG_5000_-NP-**a**, and RB-PEG-NP-**c** (RB grafted at the hydroxyl groups of MSNs, Table 2) the estimated RB loading was equal, and consequently, the observed dye aggregation in these samples should be assigned to interparticle processes, as supported by Zpot values and previously attributed to a lower presence of PEG molecules at the external surface. Sample RB-PEG-NP-**b**, with RB loading at the amine groups half of that obtained for the RB at the hydroxyl groups (sample RB-PEG-NP-**b** vs. sample RB-PEG-NP-**a** in Table 2), showed a narrower absorption band, not much different from that recorded for RB in diluted solution [105]. However, reducing the cargo of PS per nanoparticle would compel a higher concentration of nanoparticles per volume to reach effective PS doses for PDT in the cells, which would also promote particle-particle agglomeration. For this reason, the optimization of the samples is not a trivial task, and the quantification of their singlet oxygen production would be a good indicator of their applicability in cells. Significantly, all the samples, except for RB-PEG_750_-NP-**a**, showed a similar singlet oxygen quantum yield, with values around Φ_Δ_ ≈ 0.80–0.85 in deuterated methanol (CH_3_OD), similar to that registered for RB in the same solvent (Φ_Δ_ = 0.86). The fact that RB grafted to MSN can generate singlet oxygen as efficiently as the RB in solution is indicative of the potential use of these nanosystems in PDT [65,67,108].

In this context, we considered MSNs with PS and PEG at OH- groups the best nanosystems for PDT in terms of maximized PS loading with good stability in aqueous media. Nevertheless, in the series of novel in lab-made BODIPY-based PS, homologous molecular structure with carboxyl or silylated groups as graftable groups are proposed (BDP2 vs. BDP3; BDP4 vs. BDP5 and BDP6 vs. BDP7 in Figure 1) to compare their respective PDT action for a normalized concentration of PS incubated in cells. In a further step, FA was anchored through its carboxylic function to the amine groups of NH-MSN to increase the nanosystem internalization into cells, as demonstrated in a former work [105]. The presence of FA at the external surface of PS-PEG-MSNs also assists to the stability of the nanoparticles reducing the interparticle aggregation as it was experimentally verified by the absorption spectra of RB in sample RB-PEG-NP-**d** (Figure 3) in comparison with sample RB-PEG-NP-**c** without FA, and whose band shape resembles that of RB in the sample of RB-PEG-NP-**b**. The successful tethering of FA at the MSNs was checked by its characteristic band at 350 nm (Appendix A) [105,113], although its accurate quantification was not possible because of the important scatter contribution in this region.

Based on the in vitro experiments, both RB-PEG-NP-**d** (RB and PEG grafted at the OH- and FA at NH_2_-) and RB free in solution were not cytotoxic under dark conditions (Figure 4, Appendix A). When exposed to light, RB-PEG-NP-**d** showed a higher phototoxicity compared to RB in solution at the same PS concentration (Figure 4 and Appendix A). At a normalized RB concentration of 1 µM, RB-PEG-NP-**d** decreased cell viability by 80%, while RB alone in solution decreased cell viability by 50% (Figure 4). The EC_50_ value for RB-PEG-NP-**d** exposure was 0.55 µM, while in exposure to RB alone it was 1.05 µM (Appendix A). This is probably related to a higher internalization of the RB-PEG-NP-**d** compared to RB in solution as can be seen in Figure 5. Indeed, previous internalization assays of analog PEG-FA-MSNs but functionalized with a fluorescent dye (Rhodamine 101) demonstrated the capability of these nanosystems to accumulate specifically inside lysosomes of HeLa cells [105].

### 2.3. Photosensitized Silica Nanoparticles with Other Photosensitizers

Concerning the in-lab synthesized PSs, the molecular design of the novel lab-made BODIPY-PSs was based on previous studies [18,23,114,115,116]. The best choice to promote the intersystem crossing in haloBODIPY is the iodination at the 2 and 6 positions of the BODIPY core (BDP2 and BDP3), reaching singlet oxygen production ≥80% under green illumination (Appendix A). Another alternative without using halide atoms to preclude the cytotoxic effect inherent to heavy atoms [26,28,117,118,119], is based on orthogonal BODIPY dimers [120] (BDP4 and BDP5). Generally, these dyads are endowed with very intense absorption bands in the green region, as well as high singlet oxygen generation (Φ_Δ_ > 75%, Appendix A) promoted by intra-charge transfer states [121,122,123]. To shift the absorption band to the blue region (BDP1), a nitrogen atom was placed at the *meso* position of the BODIPY together with iodines at 2 and 6. The formation of a hemicyanine-like structure induced a very pronounced blue-shift, placing the main absorption band in the blue region (at around 420–440 nm) [22] but keeping a good singlet oxygen generation (≈80%, Appendix A). Most interestingly for PDT, to shift the absorption into the clinic window (650–850 nm), conjugated systems [116], particularly styryl groups with electron-donating methoxy groups in 3 and 4 positions of the phenyl ring, were added at the 3 and 5 positions, accompanied by iodine atoms at the 2 and 6 positions (BDP6 and BDP7). The singlet oxygen quantum yield achieved for these red-haloBODIPYs was lower, at around 45% (Appendix A) with respect to the other BDP-PS in this series, but they revealed an emission ability of around 20%, which can enable fluorescence imaging, in contrast to the rest of the non-emissive haloBDP (Φ_fl_ ≤ 0.03, Appendix A). Additionally, graftable groups were also incorporated in the *meso* position into all of these BODIPYs to allow their linking to MSNs.

Thus, PSs (commercial and lab-made) were classified according to their absorption range (blue, green and red) and their respective graftable groups (silylated: BDP1, BDP3, BDP5 and BDP7; carboxylic: C6, BDP2, BDP4 and BDP6; and amine group: Th, Figure 1). The outer surface of MSNs was also decorated with PEG and FA, tethering at the hydroxyl groups of MSN and at amine groups, respectively, except for the COOH-MSN, in which a modified folic acid, FA-HDA (see Materials and Methods section), was linked to the carboxylic groups.

The photophysical features, the singlet oxygen production and the phototoxic action of PS-PEG-FA-MSNs in HeLa cells were compared with their chemically homologous PSs with carboxylic group since the silylated ones polymerize in cell culture media (Figure 6, Table 3 and Appendix A).

### 2.4. In Vitro Experiments in HeLa Cells

The most representative lab-made PS-MSNs were tested in HeLa cells, applying different light sources depending on the absorption band positions: BDP1-NP (*λ*_max_ = 435 nm) under blue irradiation (*λ*_max_ = 435 nm at 10 J/cm^2^), BDP3-NP, BDP4-NP and BDP5-NP (*λ*_max_ = 510–530 nm) under green irradiation (*λ*_max_ = 518 nm at 10 J/cm^2^) and finally C6-NP and BDP6-NP (*λ*_max_ = 635–660 nm) were irradiated under red light (*λ*_max_ = 655 nm at 15 J/cm^2^). Unfortunately, none of our available irradiation sources (blue, green or red, see experimental section) were suitable to activate the Th photosensitizer (*λ*_ab_ ≈ 600 nm).

Under dark conditions, some PSs free in solution, such as halo-BODIPY, BDP2 and BDP6, as well as the commercial C6, were toxic to HeLa cells at concentrations ≥1 μM (Figure 7 and Figure 8, Appendix A). On the other hand, when these PSs were grafted to the MSNs no cytotoxicity was observed under dark conditions (Figure 7, Figure 8, Appendix A). In general terms, PS-PEG-FA-MSN did not show any toxicity under dark conditions (Appendix A), except for the BDP4-NP (Appendix A) sample with a low PS loading at the MSN (5 μmol/g, Table 3), which required a higher amount of nanoparticles (≈0.9 mg/mL) to achieve an equal PS concentration of 5 μM incubated in HeLa cells. To avoid cytotoxicity and to guarantee the safe use of PS-PEG-FA-MSN, the upper limit was set at 1 mg/mL, a value in good agreement with a former work [105].

The BDP1-NP nanosystem was able to induce around 70% cell death at 1 μM of PS concentration and near 90% cell death at 5 µm under blue light irradiation, leading to a EC_50_ = 1.0 μM (Appendix A). Better performance was revealed by the haloBDP nanosystem BDP3-NP activated by green irradiation light, which induced ≥80% cytotoxicity under light exposure at 0.5 μM PS and ≥90% at 1 μM PS (Figure 7A), providing an EC_50_ of 0.4 µM (Appendix A) without any cytotoxic effects under dark conditions. However, although the homologous haloBDP BDP2 free in solution induced higher phototoxicity under light conditions (Figure 7B, Appendix A), attributed to a higher oxygen singlet production than PS at the MSN surface (Φ_Δ_ = 0.95 vs. 0.70, Table 3 and Appendix A, respectively), it also induced cytotoxicity under dark conditions (EC_50_(DARK) ≈ 4 mM, Appendix A). Conversely, halogen-free BODIPY dimer BDP4 did not show toxicity in the dark, but its phototoxicity action was also greatly reduced (Figure 7C). However, the fact that this heavy-atom-free PS did not show dark toxicity allows a safe increase of the dimer incubated doses, reaching a 75% and 90% decrease of cell viability at 5 μM and 10 μM PS concentrations, respectively (Figure 7C).

Please note that the phototoxicity is drastically enhanced when the BODIPY dimer is loaded at the MSN surface (e.g., BDP5-NP) leading to an EC_50_ value 40 times higher (Appendix A). The Figure 6C,D show how BDP5-NP induced a decrease of 90–100% in HeLa viability at 0.5 µM, whereas nearly no phototoxicity was observed for the BODIPY dimer at the same concentration. The lower cytotoxicity effect under light irradiation of BDP4, a larger molecule with a lesser solubility in aqueous solution, could be likely assigned to a poor internalization into the cells.

These promising results demonstrated the efficiency of functionalized MSNs to successfully transport PSs into HeLa cells, promoting higher phototoxicity at a lower concentration of PSs in comparison with the PS free in solution, and avoiding cytotoxicity under dark conditions. Most importantly, the proposed BDP-PEG-FA-MSNs (both iodinated and orthogonal BODIPY dimers) showed a higher phototoxicity effect in comparison to analogous nanosystems loaded with commercial RB under the same green light doses (Figure 4 and Figure 7, and Appendix A).

The designed BDP-based PSs for red-irradiation generally displayed lower phototoxicity with respect to those in the green region (Figure 8 vs. Figure 7), which is in agreement with their lower oxygen singlet production (Table 3 and Appendix A). However, the red-BODPY-PS demonstrated a higher ability to kill cells under red light irradiation than the commercial and widely used C6, despite a lower oxygen singlet production with respect to C6. The lower oxygen singlet production is balanced with a higher absorption coefficient, reaching higher phototoxicity (Appendix A) and resulting in lower EC_50_ under red light irradiation (Appendix A), but also higher cytotoxicity under dark conditions (Figure 8A–C).

Although the photoactivity of the red-PSs loaded at MSN, samples BDP6-NP and C6-NP, with respect to PSs free in solution was inferior, once again, MSNs hampered the inherent toxicity of PSs in dark, which is of interest for clinical studies. In this context, PDT efficiency could be increased by safely applying higher concentrations of PS-MSNs and by increasing the exposure time of the irradiation [124]. Additionally, these functionalized nanosystems loaded with red-BDP PS were endowed with enough brightness to be tracked by fluorescence microscopy (Appendix A).

## 3. Materials and Methods

### 3.1. Materials and Methods

All starting materials and reagents for the MSNs synthesis were used as commercially provided unless otherwise indicated. Tetraethoxysilane (TEOS) (≥99%), ammonium hydroxide solution (NH_4_OH) (25% NH_3_ basis), hexadecyltrimethylammonium bromide (CTAB) (≥98%), 3-aminopropyltrimethoxysilane (APTMS) (97%), 3-cyanopropyltriethoxysilane (CTES) (98%), 3-aminopropyltriethoxysilane (APTES) (99%), 1-hydroxybenzotriazole hydrate (HOBt) (≥97%), boron trifluoride diethyl etherate (for synthesis), dimethylformamide anhydrous (DMF) (99.8%), 3,4-dimethoxybenzaldehyde (99%), piperidine (≥99.5%), 2,3-dichloro-5,6-dicyano-1,4-benzoquinone (DDQ) (98%), *N*-hydroxysuccimide (NHS) (98%), *N*-(3-(dimethylaminopropyl)-*N**’*-ethylcarbodiimide (EDC) (≥97%), triethylamine (TEA) (≥99%) and folic acid (FA) (≥97%) were purchased from Sigma-Aldrich (St. Louis, MO, USA); Ethyl chloroformate (≥99%) and 2,4-dimethylpyrrole (97%) were supplied by Acros (Geel, Belgium); trifluoroacetic acid (TFA) (99%) was purchased from Alfa Aesar (Haverhill, MA, USA); acetic acid glacial (synthesis grade) from Scharlab (Debrecen, Hungary); polyethylene glycol (Si-PEG) (2000 Da, >95%) from Iris BIOTECH GMBH (Maktredwitz, Germany); NHS-PEG derivative (750 Da (>95%) and 2000 Da (>95%), supplied by Iris BIOTECH GMBH, and 5000 Da (≥80%) by Sigma-Aldrich.

### 3.2. Synthesis of New BODIPY-Based PSs

#### 3.2.1. General

Anhydrous solvents were prepared by distillation over standard drying agents according to common methods. All other solvents were of HPLC grade and were used as provided. Flash chromatography was performed using silica gel (230–400 mesh). NMR spectra were recorded using CDCl_3_ or CDCl_3_/CD_3_OD at 20 °C. ^1^H NMR and ^13^C NMR chemical shifts (*δ*) were referenced to internal solvent CDCl_3_ (*δ* = 7.260 and 77.03 ppm, respectively) or CD_3_OD (*δ* = 3.205/4.031 and 52.69 ppm, respectively). Multiplicity is indicated as follows: s = singlet; d = doublet; dd = double doublet; t = triplet; q = quadruplet; quint = quintuplet; m = multiplet. Coupling constants (*J*) are dated in hertz (Hz). DEPT 135 experiments were used to determine the type of carbon nucleus (C vs. CH vs. CH_2_ vs. CH_3_). FTIR spectra were obtained from neat samples using the attenuated total reflection (ATR) technique. High-resolution mass spectrometry (HRMS) was performed using electronic impact (EI) or MALDI-TOF and ion trap (positive mode) for the detection.

BDP1 [109], BDP2 [125], 8-(4-carboxyphenyl)-2-formyl-1,3,5,7-tetramethylBODIPY [126] and 8-(4-carboxyphenyl)-2,6-diiodo-1,3,5,7-tetramethylBODIPY [127] were synthesized by the corresponding described methods. The synthesis of BODIPYs BDP3-BDP7 is illustrated in Appendix A.

#### 3.2.2. General Procedure for the Formation of Amides

The corresponding carboxy-BODIPY (1 mol. equiv.), APTES (2.1 mol. Equiv.), TEA (2 mol. equiv.), EDC (2 mol. equiv.) and HOBt (2 mol. equiv.) were dissolved in CH_2_Cl_2_ and stirred under argon at rt for 12 h. The reaction mixture was then washed with HCl 10% and water. The obtained organic layer was dried over anhydrous Na_2_SO_4_, filtered and the solvent evaporated to dryness. The obtained residue was submitted to purification by flash chromatography on silica gel.

#### 3.2.3. Synthesis of BDP3

According to the general procedure described in Section 3.2.2., BDP2 [125] (70 mg, 0.12 mmol), APTES (0.06 mL, 0.25 mmol), TEA (0.03 mL, 0.24 mmol), EDC (46 mg, 0.24 mmol) and HOBt (32 mg, 0.24 mmol) in CH_2_Cl_2_ (10 mL) were reacted. Flash chromatography (CH_2_Cl_2_/EtOAc, 90:10) afforded BDP3 (39 mg, 41%) as an orange-red solid. ^1^H NMR (300 MHz, CDCl_3_) *δ* 5.84 (t, *J* = 5.4 Hz, 1H, NH), 3.82 (q, *J* = 6.9 Hz, 6H, 3CH_2_O), 3.26 (q, *J* = 6.9 Hz, 2H, CH_2_N), 3.11–3.05 (m, 2H, CH_2_), 2.61 (s, 6H, 2CH_3_), 2.49 (s, 6H, 2CH_3_), 2.32 (t, *J* = 6.9 Hz, 2H, CH_2_), 1.99–1.90 (m, 2H, CH_2_), 1.64 (quint, *J* = 7.8 Hz, 2H, CH_2_), 1.22 (t, *J* = 6.9 Hz, 9H, 3CH_3_), 0.64 (t, *J* = 7.8 Hz, 2H, CH_2_Si) ppm. ^13^C NMR (75 MHz, CDCl_3_) *δ* 171.1 (CONH), 155.5 (C), 145.3 (C), 142.5 (C), 131.5 (C), 86.6 (C-I), 58.5 (CH_2_), 41.9 (CH_2_), 36.1 (CH_2_), 28.4 (CH_2_), 27.3 (CH_2_), 22.9 (CH_2_), 19.0 (CH_3_), 18.3 (CH_3_), 16.2 (CH_3_), 7.9 (CH_2_) ppm. FTIR *ν* 3302, 2971, 2924, 1712, 1620, 1542, 1463, 1392, 1346, 1190, 1084, 1003, 958 cm^−1^. HRMS-EI *m/z* 789.0931 (calcd. for C_26_H_40_BF_2_I_2_N_3_O_4_Si: 789.0939).

#### 3.2.4. Synthesis of BDP4

To a degassed solution of 8-(4-carboxyphenyl)-2-formyl-1,3,5,7-tetramethylBODIPY [126] (186 mg, 0.47 mmol) in dry CH_2_Cl_2_ (15 mL) were added a solution of 2,4-dimethylpyrrole (0.10 mL, 0.98 mmol) in dry CH_2_Cl_2_ (2 mL) and two drops of TFA, and the resulting mixture stirred at rt for 2 h. After, a solution of DDQ (117.0 mg, 0.51 mmol) in CH_2_Cl_2_ (10 mL) was added, and the resulting new mixture stirred for 30 min. Then, TEA (0.32 mL, 2.34 mmol) and BF_3_ Et_2_O (0.58 mL, 4.68 mmol) were added to the mixture, and the resulting final mixture stirred for 3 h at rt, washed with HCl 10%, and water. The obtained organic layer was dried over anhydrous Na_2_SO_4_, filtered and the solvent evaporated to dryness. Flash chromatography (CH_2_Cl_2_/EtOAc, 30:70) afforded BDP4 (78 mg, 27%) as an orange solid. ^1^H NMR (300 MHz, CDCl_3_) *δ* 8.27 (d, *J* = 8.1 Hz, 2H, 2CH), 7.46 (d, *J* = 8.1 Hz, 2H, 2CH), 6.10 (s, 1H, CH), 5.99 (s, 2H, 2CH), 2.61 (s, 3H, CH_3_), 2.52 (s, 6H, 2CH_3_), 2.42 (s, 3H, CH_3_), 1.71 (s, 6H, 2CH_3_), 1.41 (s, 3H, CH_3_), 1.21 (s, 3H, CH_3_) ppm. ^13^C NMR (75 MHz, CDCl_3_) *δ* 170.6 (COOH), 159.4 (C), 155.9 (C), 151.3 (C), 145.2 (C), 142.3 (C), 140.7 (C), 140.2 (C), 138.0 (C), 133.2 (C), 132.0 (C), 131.8 (C), 131.2 (CH), 130.4 (C), 130.3 (C), 128.4 (CH), 126.0 (C), 122.9 (CH), 121.3 (CH), 14.9 (CH_3_), 14.8 (CH_3_), 14.6 (CH_3_), 14.0 (CH_3_), 12.8 (CH_3_), 12.3 (CH_3_) ppm. FTIR *ν* 2924, 2855, 1709, 1546, 1464, 1311, 1193, 1078, 981 cm^−1^. HRMS-EI *m/z* 614.2636 (calcd. for C_33_H_32_B_2_F_4_N_4_O_2_: 614.2648).

#### 3.2.5. Synthesis of BDP5

According to the general procedure described in Section 3.2.2, BDP4 (46 mg, 0.07 mmol), APTES (0.04 mL, 0.15 mmol), TEA (0.02 mL, 0.14 mmol), EDC (28 mg, 0.14 mmol) and HOBt (20 mg, 0.15 mmol) in CH_2_Cl_2_ (10 mL) were reacted. Flash chromatography (hexane/EtOAc, 60:40) afforded BDP5 (30 mg, 49%) as an orange solid. ^1^H NMR (300 MHz, CDCl_3_) *δ* 7.95 (d, *J* = 8.1 Hz, 2H, 2CH), 7.39 (d, *J* = 8.1 Hz, 2H, 2CH), 6.71 (t, *J* = 5.4 Hz, 1H, NH), 6.08 (s, 1H, CH), 5.98 (s, 2H, 2CH), 3.82 (q, *J* = 6.9 Hz, 6H, 3CH_2_O), 3.50 (q, *J* = 6.0 Hz, 2H, CH_2_N), 2.60 (s, 3H, CH_3_), 2.52 (s, 6H, 2CH_3_), 2.41 (s, 3H, CH_3_), 1.79 (quint, *J* = 7.8 Hz, 2H, CH_2_), 1.70 (s, 6H, 2CH_3_), 1.39 (s, 3H, CH_3_), 1.21 (t, *J* = 6.9 Hz, 9H, 3CH_3_), 1.20 (s, 3H, CH_3_), 0.73 (t, *J* = 7.8 Hz, 2H, CH_2_Si) ppm. ^13^C NMR (75 MHz, CDCl_3_) *δ* 166.3 (CONH), 159.2 (C), 155.8 (C), 151.1 (C), 145.3 (C), 142.3 (C), 141.1 (C), 138.1 (C), 137.6 (C), 135.8 (C), 133.3 (C), 132.2 (C), 131.8 (C), 130.5 (C), 128.2 (CH), 128.1 (CH), 125.9 (C), 122.8 (CH), 121.3 (CH), 58.6 (CH_2_O), 42.4 (CH_2_), 22.8 (CH_2_), 18.3 (CH_3_), 14.83 (CH_3_), 14.80 (CH_3_), 14.6 (CH_3_), 13.9 (CH_3_), 12.8 (CH_3_), 12.3 (CH_3_), 8.0 (CH_2_) ppm. FTIR *ν* 2924, 2854, 1646, 1546, 1513, 1310, 1193, 1078, 980 cm^−1^. HRMS-MALDI-TOF *m/z* 817.3979 (calcd. for C_42_H_53_B_2_F_4_N_5_O_4_Si: 817.3989).

#### 3.2.6. Synthesis of BDP6

8-(4-Carboxyphenyl)-2,6-diiodo-1,3,5,7-tetramethylBODIPY [127] (25 mg, 0.04 mmol) in DMF (2 mL), 3,4-dimethoxybenzaldehyde (20 mg, 0.12 mmol), piperidine (0.02 mL, 0.20 mmol) and acetic acid (0.01 mL, 0.20 mmol) were added to a microwave tube. The tube was sealed with an aluminum cap and heated for 20 min at 80 °C under microwave radiation (Biotage^®^ Initiator Classic, Uppsala, Sweden). After being cooled down to rt, CH_2_Cl_2_ was added and the organic layer washed with water, dried over anhydrous Na_2_SO_4_, filtered, and evaporated to dryness. The obtained residue was submitted to purification by flash chromatography on silica gel with EtOAc/CH_3_OH (95:5) to give BDP6 (16 mg, 43%) as a green solid. ^1^H NMR (700 MHz, CDCl_3_/CD_3_OD 4:1) *δ* 8.09 (d, *J* = 8.4 Hz, 2H, 2CH), 7.98 (d, *J* = 16.8 Hz, 2H, 2CH=C), 7.42 (d, *J* = 16.8 Hz, 2H, 2C=CH), 7.30 (d, *J* = 8.4 Hz, 2H, 2CH), 7.09 (dd, *J* = 8.4 and 2.1 Hz, 2H, 2CH), 7.04 (d, *J* = 2.1 Hz, 2H, 2CH), 6.78 (d, *J* = 8.4 Hz, 2H, 2CH), 3.82 (s, 6H, 2CH_3_O), 3.79 (s, 6H, 2CH_3_O), 1.31 (s, 6H, 2CH_3_) ppm. ^13^C NMR (176 MHz, CDCl_3_/CD_3_OD 4:1) *δ* 172.0 (COOH), 154.7 (C), 154.4 (C), 153.1 (C), 149.5 (C), 143.7 (C), 143.6 (CH), 143.5 (CH), 141.1 (C), 136.4 (C), 135.9 (C), 134.8 (CH), 134.7 (CH), 133.8 (C), 132.7 (CH), 132.6 (CH), 125.8 (CH), 120.7 (CH), 115.2 (CH), 115.1 (CH), 113.7 (CH), 87.1 (C-I), 59.82 (CH_3_O), 59.78 (CH_3_O), 21.5 (CH_3_) ppm. FTIR *ν* 2924, 2852, 1694, 1591, 1514, 1461, 1265, 1176, 1101, 1012 cm^−1^. HRMS-MALDI-TOF *m/z* 906.0638 (calcd. for C_37_H_35_BF_2_I_2_N_2_O_6_: 906.0646).

#### 3.2.7. Synthesis of BDP7

According to the general procedure described in Section 3.2.2, BDP6 (37 mg, 0.04 mmol), APTES (0.02 mL, 0.085 mmol), TEA (0.01 mL, 0.08 mmol), EDC (15 mg, 0.08 mmol) and HOBt (11 mg, 0.08 mmol) in CH_2_Cl_2_ (10 mL) were reacted for 12 h. Flash chromatography (CH_2_Cl_2_/EtOAc, 90:10) afforded BDP7 (14 mg, 32%) as a red solid. ^1^H NMR (700 MHz, CDCl_3_) *δ* 8.12 (d, *J* = 16.8 Hz, 2H, 2CH=C), 7.99 (d, *J* = 8.4 Hz, 2H, 2CH), 7.57 (d, *J* = 16.8 Hz, 2H, 2C=CH), 7.41 (d, *J* = 8.4 Hz, 2H, 2CH), 7.23 (dd, *J* = 8.4 and 1.4 Hz, 2H, 2CH), 7.16 (d, *J* = 1.4 Hz, 2H, 2CH), 6.90 (d, *J* = 8.4 Hz, 2H, 2CH), 6.79 (t, *J* = 6.3 Hz, 1H, NH), 3.96 (s, 6H, 2CH_3_O), 3.93 (s, 6H, 2CH_3_O), 3.86 (q, *J* = 7.0 Hz, 6H, 3CH_2_O), 3.54 (q, *J* = 6.3 Hz, 2H, CH_2_N), 1.85–1.80 (m, 2H, CH_2_), 1.43 (s, 6H, 2CH_3_), 1.24 (t, *J* = 7.0 Hz, 9H, 3CH_3_), 0.77 (t, *J* = 7.7 Hz, 2H, CH_2_Si) ppm. ^13^C NMR (176 MHz, CDCl_3_) *δ* 166.4 (CONH), 150.7 (C), 150.5 (C), 149.2 (C), 145.4 (C), 139.6 (CH), 138.4 (C), 137.0 (C), 135.9 (C), 132.6 (C), 129.8 (C), 128.8 (CH), 128.1 (CH), 121.9 (CH), 116.9 (CH), 111.1 (CH), 109.7 (CH), 83.3 (C-I), 58.6 (CH_2_O), 56.1 (CH_3_O), 56.0 (CH_3_O), 42.4 (CH_2_), 22.8 (CH_2_), 18.4 (CH_3_), 17.8 (CH_3_), 8.0 (CH_2_) ppm. FTIR *ν* 3487, 2921, 2852, 1743, 1514, 1463, 1176, 1099, 1014 cm^−1^. HRMS-MALDI-TOF *m/z* 1119.1824 (calcd. for C_47_H_54_BF_2_I_2_N_3_O_8_Si: 1119.1831).

### 3.3. Synthesis of the MSNs

The synthesis of mesoporous silica nanoparticles with amine groups at the external surface (NH-MSNs) has been described previously [105,109]. The synthesis route of mesoporous silica nanoparticles with carboxylic group (COOH-MSN) is similar to that described for NH-MSN, but in the second step, CTES (0.007 mmol) is added instead of APTMS to provide cyane groups at the external surface. After collection, CN-MSNs were re-suspended in acid solution (H_2_O:H_2_SO_4_ 50:50 *v/v*) and stirred under 140 °C for 12 h to convert the CN into COOH groups. Then, the sample was cooled down to rt and kept under stirring for another 24 h. Finally, the COOH-MSNs were washed several times with water, until neutral supernatants were collected, Appendix A.

### 3.4. Grafting of Molecules (PS, PEG and FA) on the MSN Surface

The different functionalized MSNs, varying PS, PEG and FA (Figure 1) and the linkage approach, were synthesized and named as follows:-RB-PEG-NP-**a**: RB (0.03 mmol), previously silylated with an equimolar ratio of APTMS (0.03 mmol) in 20 mL of CH_3_CN under stirring for 1 h under inert atmosphere [109], was directly coupled to the structural hydroxyl groups of MSN (40 mg) added afterward. The reaction was kept for 3 h and the nanoparticles were collected by filtration. Then, RB-MSNs were re-suspended in 20 mL of CH_3_CN and then, NHS ester-activated PEG (0.03 mmol), NHS-PEG (Figure 1), was added to react for 3 h with the external primary amine group of NH-MSN to yield stable amide bonds and releasing *N*-hydroxysuccinimide group (NHS). In this type of MSN, the pelygation procedure of the external surface was carried out with three different PEG chains of 750, 2000, and 5000 Da. The corresponding samples were named RB-PEG_750_-NP-**a**, RB-PEG_2000_-NP-**a** and RB-PEG_5000_-NP-**a**, respectively.-RB-PEG-NP-**b**: RB was linked through its carboxylic group to the amine groups of NH-MSN (40 mg) by the carbodiimide method. This method has been previously described [105,109]. Briefly, NHS/EDC is added to RB in solution (0.03 mmol in 20 mL of CH_3_CN) in equimolar concentration and stirring for 1 h in an inert atmosphere. Then NH-MSN (40 mg) was directly added to the reaction mixture and kept stirring for 3 h, and nanoparticles were collected. In a second step, PEG (0.03 mmol) with a silylated group at one edge (Si-PEG, Figure 1) was externally anchored to the inherent hydroxyl groups of RB-MSN (40 mg) by direct condensation reaction during 3 h.-RB-PEG-NP-**c**: both RB (0.03 mmol), previously silylated according to the process previously described in sample RB-PEG-NP-**a**, and Si-PEG (0.03 mmol) were simultaneously bound in CH_3_CN to the external hydroxyl groups of MSN (40 mg) following the procedure previously mentioned.-RB-NP-**d**: FA (0.03 mg) was added to RB-PEG-NP-**c** (40 mg), previously re-suspended in CH_3_CN (20 mL), and was linked to the amine groups of RB-PEG-NP-**c** sample through the carbodiimide method cited above.

For the rest of the PDT-nanosystems, denoted as PS-NP, with PS being the short name of every PS (indicated in Figure 1), the linkage between PS and MSN is ruled out by the main functional group at the PS (Figure 1). That is, the amine group of Th dye is covalently linked to the carboxylic groups of COOH-MSN (previously treated with NHS/EDC). Photosensitizers with silylated groups (BDP1, BDP3, BDP5 and BDP7) were directly coupled to the hydroxyl groups of OH-MSN. Finally, PSs with carboxylic groups (C6, BDP2, BDP4, and BDP6) were grafted to the external amine groups of nanoparticles (NH-MSNs). Note here that C6 was linked by the previous carbodiimide methods but BODIPYs (BDP2, BDP4, and BDP6) cannot withstand such conditions, and an alternative synthetic route was employed. BODIPY-COOH was dissolved in CH_3_CN anhydride (20 mL) at 0 °C, ethyl chloroformate and triethylamine were added dropwise in equimolar concentration (0.03 mmol) and the system was kept under vigorously stirring for 30 min. Then, NH-MSNs were added at rt and stirred for other 30 min. Finally, the functionalized nanoparticles were washed with EtOH until a colorless supernatant was obtained.

In all these samples, Si-PEG (2000 Da) was tethered at the hydroxyl groups and FA to the amine groups of MSNs. The exception was Th-MSN, in which FA was linked to carboxyl groups by a previous chemical modification of FA, according to Appendix A [101]. Briefly, the terminal COOH group of FA was modified with *N*-Boc-1,6-hexanediamine (Boc-HDA) and the amine group (FA-HDA) was obtained after removing the Boc group, according to [101]. The relative quantity of PS, PEG and FA added to the reaction per mg of nanoparticles was equivalent for all PS-MSN samples.

### 3.5. Physicochemical and Photophysical Characterization

The size, shape and morphology of the silica nanoparticles were characterized by electron microscopes, scanning electron microscopy (SEM) and transmission electron microscopy (TEM). SEM images were obtained using a JEOL JSM-6400 (JEOL, Tokyo, Japan) and TEM images were obtained using a Philips SuperTwin CM200 (Thermo Fisher Scientific, Eindhoven, The Netherlands) at 200 kV. The nanoparticle size distribution was analyzed by Image-J software (1.52u, National Institute of Health, Bethesda, MD, USA). Dynamic light scattering (DLS) and Zeta potential (Zpot) measurements to analyze the NP size and their stability in suspension were carried out using a Malvern Zetasizer Nano ZS (Malvern Products, Madrid, Spain), which had a Helio-Neon laser (*λ* = 633 nm). FTIR spectra were obtained from neat samples in powder using ATR technique in an Anity-1S Shimadzu spectrometer (Izasa Scientific, Barcelona, Spain) (4000–400 cm^−1^ range) and XPS spectra were recorded using a SPECS system (Berlin, Germany) with a Phoibos 150 1D-DLD analyzer and Al Kα (1486.7 eV) as monochromatic radiation.

The absorption spectra were recorded by UV-Vis-NIR Spectroscopy (model Cary 7000, Agilent Technologies, Madrid, Spain) equipped with two lamps (halogen lamp for Vis-IR region and deuterium lamp for UV region). In the case of the silica nanoparticle samples, an integrating sphere (model Internal DRA 900, Livingston, UK) was used to reduce the scatter of the samples. The fluorescence measurements were recorded with an Edinburgh Instruments Spectrofluorimeter (FLSP920 model, Livingston, UK) equipped with a xenon flash lamp 450 W as the excitation source. The fluorescence spectra were corrected from the wavelength dependence on the detector sensibility.

The fluorescence quantum yields of the photosensitizers were measured by the relative method, and different standard samples depending on the spectral region: coumarin 152 (Φ_fl_ = 0.19 in EtOH) for the blue region [128], PM597 (Φ_fl_ = 0.32 in cHex) [20] for the green and cresyl violet (Φ_fl_ = 0.54 in CH_3_OH) [129] and zinc phthalocyanine (Φ_fl_ = 0.30 in 1% pyridine in toluene) [130] for the red-visible region.

Radiative decay curves were recorded in the same Edinburgh Instrument by Time-Correlated Single-Photon Counting Technique (TC-SPC), using a microchannel plate detector (Hamamatsu C4878) with picoseconds time resolution (≈20 ps). Fluorescence decay curves were monitored at the maximum emission wavelength after excitation by means of a fianium supercontinuous wavelength tunable laser with 150 ps FWHM pulses.

The singlet oxygen (^1^O_2_) production was determined by direct measurement of their phosphorescence at 1276 nm employing NIR detector (InGaAs detector, Hamamatsu G8605-23), integrated into the same Edinburgh spectrofluorimeter upon continuous monochromatic excitation (450 W Xenon lamp) of the sample. Singlet oxygen quantum yields (Φ_Δ_^PS^) were calculated by the relative method, using commercial photosensitizers as references: Rose Bengal (RB, Φ_Δ_^PS^ = 0.86 in CH_3_OD), MeSBDP (CAS-1835282-63-7, Φ_Δ_^PS^ = 0.98 in CH_3_OD and Φ_Δ_^PS^ = 0.91 in CHCl_3_) [131] and New Methylene Blue (NMB, Φ_Δ_^PS^ = 0.76 in CH_3_OD).

The amount of PSs in MSN was estimated photometrically, by reading the absorbance value of a previously weighed amount of nanoparticles in a stable suspension and assuming that the molar extinction coefficient of the dye was the same in solution as when grafted on the nanoparticles.

### 3.6. In Vitro Assays

#### 3.6.1. Cell Culture

Human cervical adenocarcinoma cells (HeLa cells, CCL2) purchased from ATCC were grown in Dulbecco’s modified Eagle’s medium (DMEM) supplemented with 10% (*v/v*) fetal bovine serum (FBS) and 50 U/mL penicillin and 50 mg/mL streptomycin, in a humidified 5% CO_2_ cells incubator at 37 °C. For the cytotoxicity study, cells were grown to monolayer confluency in 96-well microplates. For the internalization and subcellular localization study, cells were seeded in glass-bottom 35 mm Petri dishes and subconfluent monolayers were used.

#### 3.6.2. Sample Preparation and In Vitro Exposures

PS-MSNs were suspended in PBS buffer (1·10^−4^ M) and stirred for at least 24 h before the exposures. PSs alone were directly dissolved in DMSO (1·10^−3^ M). For the in vitro exposures, cells were incubated for 24 h with 1, 5, 10, 50 and 100·10^−7^ M, final concentration of each PS in solution or tethered at MSN nanosystem, in 10% FBS cell culture medium. After 24 h exposure, cells were washed three times with serum-free culture medium and maintained in the culture medium during irradiation (<30 min, depending on the fluence rate of each light source) and post-treatment time (24 h).

Irradiations were performed using light-emitting diode (LED) devices: LED Par 64 Short Q4-18 (Showtec, Burgebrach, Holland) for blue (*λ*_ab_ 455 nm) and green (*λ*_ab_ 518 nm) light and LED 36 W (KINGBO LED) for red (*λ*_ab_ 655 nm), using a total light dosage (TLD) between 10 and 15 Jcm^−2^, as it is shown in the Appendix A. The irradiation time, being in all the cases shorter than 30 min, depends on the fluence rate of each LED according to the following equation:TLD (J/cm^2^) = fluence rate (mW/cm^2^) × treatment time (s)

Different LEDs were chosen depending on the main absorption band of each sample. Parallel experiments were carried out by incubating the cells with each PS or PS-MSN nanoplatforms without irradiation to test their toxicity in dark conditions. Unexposed cells and cells exposed to 1% DMSO or MSN alone were used as controls. Four replicates of each treatment were used, and experiments were repeated three times.

#### 3.6.3. Confocal Microscopy

To evaluate internalization and subcellular localization of selected PS-MSN samples through confocal microscopy, cells were incubated for 24 h with 1 μM of RB, RB-PEG-NP-d or BDP6-NP in 10% FBS-supplemented DMEM culture medium. Unexposed cells were used as control. After exposures, cells were washed three times with culture medium and fixed with 0.4% paraformaldehyde for 10 min at 4 °C. Cells were then washed three times with culture medium and observed under an Olympus Fluorview FV500 confocal microscope (Hamburg, Germany). Fluorescence images were obtained under 561 nm excitation and 565–616 nm emission range (Alexa 568) for RB samples and at 640 nm excitation and 645–700 nm emission range for BDP-6 sample. Both fluorescence and brightfield images were obtained at the same depth. Images were edited using Fiji software (ImageJ 1.49a, National Institutes of Health, Bethesda, MD, USA).

#### 3.6.4. Cell Viability (MTT) Assay

Dark and phototoxicity were assessed in HeLa cells using the thiazolyl blue tetrazolium bromide (MTT) assay following the manufacturer’s instructions. After exposures, cells were incubated with a 50 µg/mL MTT solution for 3 h at 37 °C. Then, reduced formazan product was extracted from cells with DMSO and the absorbance was measured at 570 nm in a Biotek EL 312 microplate spectrophotometer reader (Winooski, VT, USA). Cell viability was expressed as the percentage with respect to control cells. Differences between unexposed and treated cells were analyzed through the Kruskal-Wallis test followed by the Dunn’s post hoc test. Differences between dark and light exposures at the same concentrations were analyzed through Mann-Whitney U test. EC_50_ values were calculated using the Probit test. All statistical analyses were performed using the SPSS 23.0 software (Chicago, IL, USA). Significance level was globally established at 5% (*p* < 0.05).

## 4. Conclusions

Photosensitized silica nanoparticles functionalized with PEG and FA proved to be suitable and biocompatible nanosystems able to overcome some of the drawbacks of PS, as follows: (i) avoidance of cytotoxicity under dark conditions, ensuring safe use in clinical trials for cancer treatment; (ii) increased internalization, providing a better PDT performance in HeLa cells in comparison with PSs free in solution. Overall, photosensitized silica nanoparticles with BODIPY-based PSs showed higher phototoxicity compared to commercial PSs (i.e., RB-NP vs. BDP3-NP or BDP5-NP under green light and C6-NP vs. BDP6-NP under red light irradiation). Finally, BDP-PEG-FA-MSN systems under red light irradiation also enabled fluorescence bioimaging, making them promising platforms to be implemented in PDT.

## Figures and Tables

**Figure 1 ijms-22-06618-f001:**
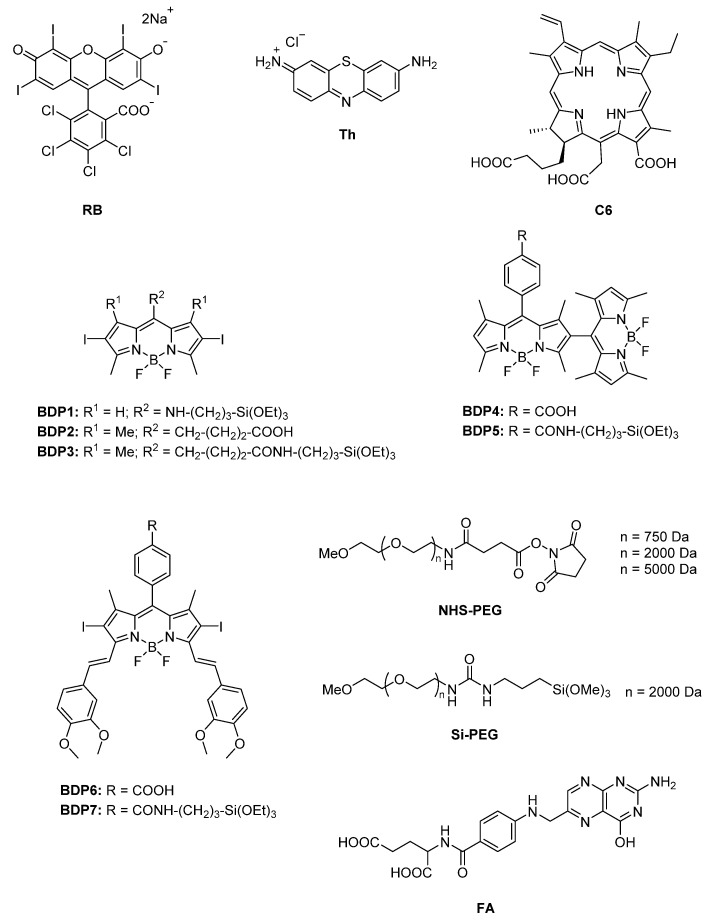
Molecular structure of the different compounds anchored to MSN: commercial (RB, Th, C6) and custom-made BODIPY photosensitizers (BDP1-BDP7), PEG derivatives with different functional groups (Si-PEG and NHS-PEG) and molecular weight (750 Da, 2000 Da and 5000 Da), and FA.

**Figure 2 ijms-22-06618-f002:**
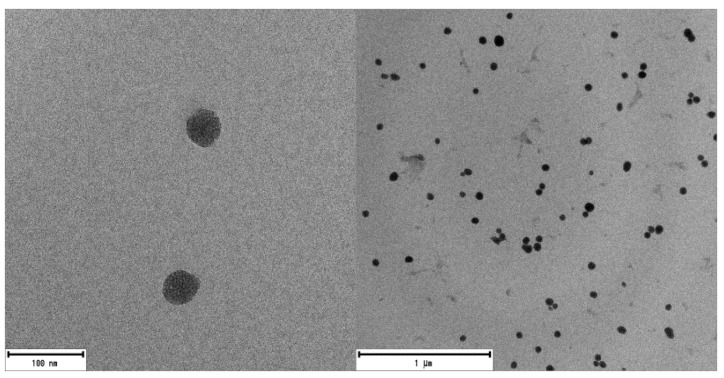
TEM images of MSNs. There are no noticeable differences between any of the synthesized MSNs (NH-MSN, CN-MSN and COOH-MSN).

**Figure 3 ijms-22-06618-f003:**
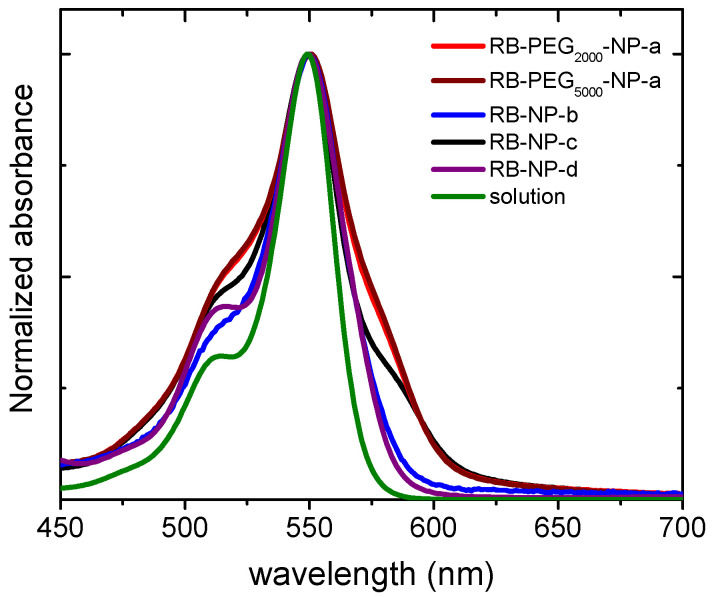
Normalized absorption spectra of RB-PEG_2000_-NP-**a** (**red**), RB-PEG_5000_-NP-**a** (**brown**), RB-PEG-NP-**b** (**blue**), RB-PEG-NP-**c** (**black**), RB-PEG-NP-**d** (**purple**) in water suspension (0.5 mg/mL) and RB in diluted aqueous solution (**green**). The absorption spectra were recorded after stirring the nanosystems for at least 24 h.

**Figure 4 ijms-22-06618-f004:**
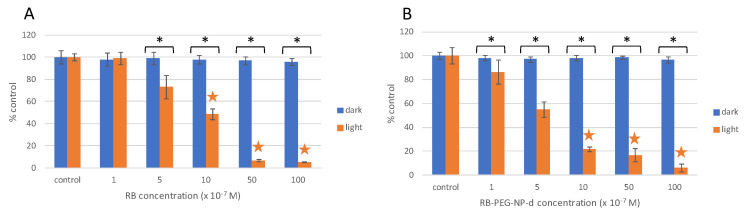
Cell viability (MTT assay) of HeLa cells exposed to different RB concentrations, (**A**) in solution, and (**B**) tethered at MNS (sample RB-PEG-NP-**d**) under dark conditions (blue bars) and green irradiation at 518 nm and 10 J/cm^2^ (orange bars). Stars indicate significant differences with respect to controls. Asterisks indicate significant differences between dark and light conditions at the same concentrations tested.

**Figure 5 ijms-22-06618-f005:**
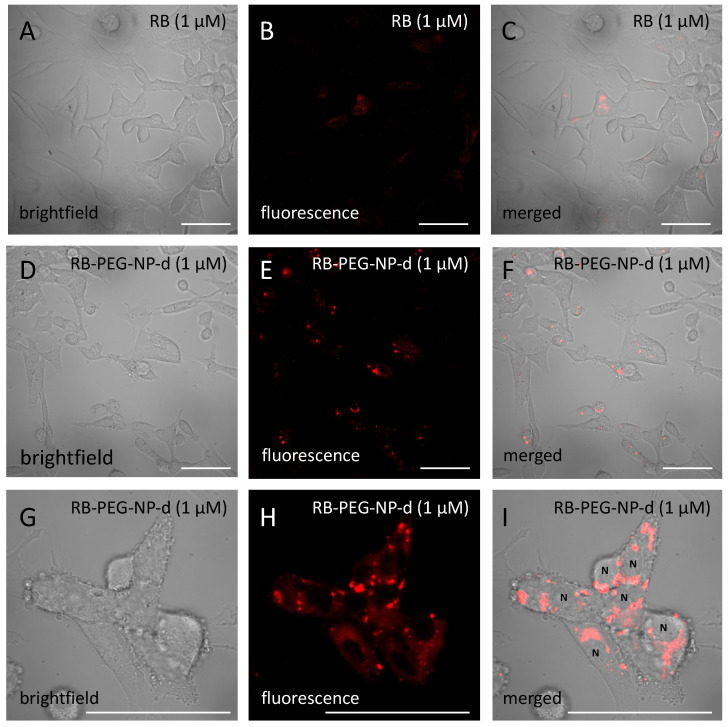
Confocal fluorescence microscopy images (*λ*ex = 561 nm and *λ*em = 565–615 nm) of HeLa cells exposed to RB free in solution (**A**–**C**) and cells exposed to RB-PEG-NP-**d** (**D**–**I**) at the same RB concentration (1 μM). Scale bars = 100 μm.

**Figure 6 ijms-22-06618-f006:**
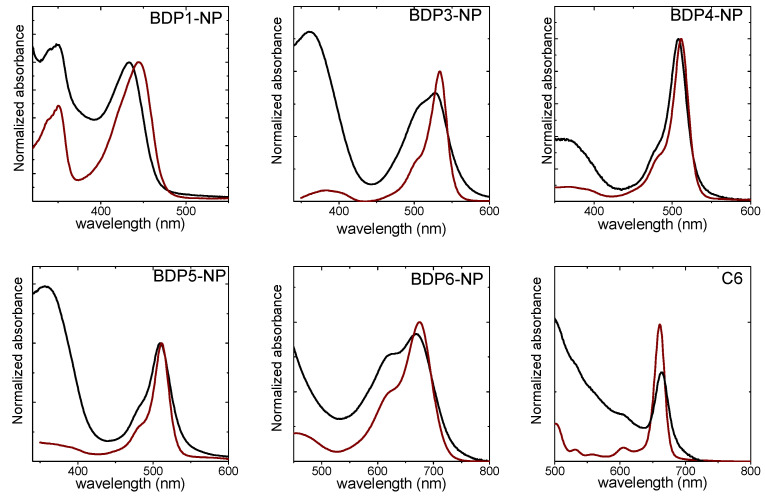
Normalized absorption spectra of PS in CHCl_3_ solution (brown) and the PS tethered at the external surface of MSN together with PEG and FA (black) in CH_3_OH at 0.5 mg/mL. The absorption spectra for all the PS-MSN samples were recorded after stirring the nanosystems for at least 24 h.

**Figure 7 ijms-22-06618-f007:**
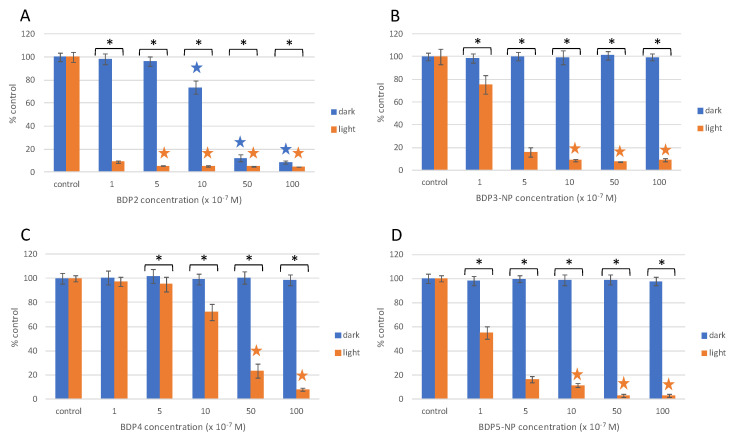
Cell viability (MTT assay) of HeLa cells exposed to the PSs in solution BDP2 (**A**) and BDP4 (**C**) and to their corresponding nanosystems BDP3-NP (**B**) and BDP5-NP (**D**) under dark conditions (blue bars) and after green irradiation at 10 J/cm^2^ (orange bars). Stars indicate significant differences with respect to controls. Asterisks indicate significant differences between dark and light conditions at the same concentrations tested.

**Figure 8 ijms-22-06618-f008:**
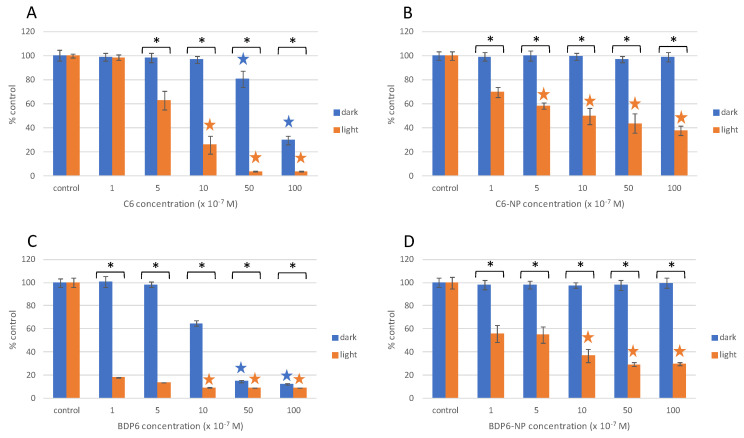
Cell viability (MTT assay) of HeLa cells exposed to PSs in solution C6 (**A**) and BDP6 (**C**) and to their respective the nanosystems C6-NP (**B**), and BDP6-NP (**D**) under dark conditions (blue bars) and after red irradiation at 15 J/cm^2^ (orange bars). Stars indicate significant differences with respect to controls. Asterisks indicate significant differences between dark and light conditions at the same concentrations tested.

**Table 1 ijms-22-06618-t001:** DLS and Zeta potential of mesoporous silica nanoparticles in water.

Name	Shell	DLS (nm)	Z Pot (mV)
NH-MSN	NH_2_/OH	71	−3.96
CN-MSN	CN/OH	280	−7.06
COOH-MSN	COOH/OH	66	−39.7

**Table 2 ijms-22-06618-t002:** RB amount, nanoparticle size and their Zeta potential by DLS in water of the different RB-PEG-MSNs.

System	Characteristic	PEG Length (Da)	DLS Size (nm)	ZPOT (mV)	[RB] (μmol/g)
RB-PEG_750_-NP-a	RB-OH-MSN PEG-NH_2_-MSN	750	130	−4.3	20
RB-PEG_2000_-NP-a	RB-OH-MSN PEG-NH_2_-MSN	2000	99	−25.0	20
RB-PEG_5000_-NP-a	RB-OH-MSN PEG-NH_2_-MSN	5000	114	−25.0	20
RB-PEG-NP-b	RB-NH_2_-MSN PEG-OH-MSN	2000	95	−29.0	10
RB-PEG-NP-c	RB-OH-MSN PEG-OH-MSN	2000	88	−31.0	20

**Table 3 ijms-22-06618-t003:** Cargo of PS at MSNs, absorption maxima (*λ*_ab_) and the singlet oxygen quantum yield in CH_3_OD (Φ_Δ_) measured after stirring the suspensions for 24 h of the PS-PEG-FA-MSN systems.

System	Characteristic	[PS] (μmol/g)	*λ*_ab_ (nm)	Φ_Δ_
BDP1-NP	BDP1-OH-MSN PEG-OH-MSN FA-NH_2_-MNS	30	435.0	0.62
BDP2-NP	BDP2-NH_2_-MSN PEG-OH-MSN FA-NH_2_-MNS	3	527.0	-
BDP3-NP	BDP3-OH-MSN PEG-OH-MSN FA-NH_2_-MNS	40	528.0	0.69
BDP4-NP	BDP4-NH_2_-MSN PEG-OH-MSN FA-NH_2_-MNS	5	513.0	0.81
BDP5-NP	BDP5-OH-MSN PEG-OH-MSN FA-NH_2_-MNS	11	511.0	0.73
BDP6-NP	BDP6-NH_2_-MSN PEG-OH-MSN FA-NH_2_-MNS	7	670.0	0.50
BDP7-NP	BDP7-OH-MSN PEG-OH-MSN FA-NH_2_-MNS	3	669.5	-
C6-NP	C6-NH_2_-MSN PEG-OH-MSN FA-NH_2_-MNS	6	662.0	0.82
Th-NP	Th-COOH-MSN PEG-OH-MSN FA-HDA-COOH-MNS	15	599.0	0.84

## Data Availability

The source data underlying Figures are available from the authors upon request.

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
