# Peer review of "Functionalization of Photosensitized Silica Nanoparticles for Advanced Photodynamic Therapy of Cancer"

_ijms, 2021, doi:10.3390/ijms22126618_

Round 1

Reviewer 1 Report

389
The fluence rate (mW/cm2) and some characteristic treatment times (s) should be given to see the progress of the total light dosage (TLD) calculation and to make the calculation process comparable to other jobs. 

394
The scheme or description of the geometric arrangement of the light irradiation should be given so that they can be compared to the cell irradiation arrangement of others.

467
The lowest zeta potential here is -37. Therefore, either the absolute value of the zeta potentials should be used in the description, or if an absolute zeta potential is used, the sentence should be reworded accordingly to correct the wording.

487
"The stability of the nanoparticles can also be studied by the absorption spectra of the RB-PEG-MSNs samples in water suspension (Figure 2)."
It should be explained that statement how the stability of the compounds can be deduced from absorption experiments. It requires a more detailed explanation and relevant references (a textbook, a review, or several explanatory articles). Especially it would be necessary since, in the figure, the RB is not in the water, as the statement indicates, but in methanol.

324
There is no section attributed to the confocal microscopy applied in the materials and methods. Please, in the manuscript, describe the confocal imaging conditions used.

541
If confocal images were made, please indicate the depth of the image sections in micrometers. If confocality was not applied, please indicate it appropriately here.

544
The results and discussion section would be better if the parts would be emphasized with subtitles and placed in separate paragraphs. In its current form, experiment and statements are very converging. The reader's attention would be better guided if the more important parts were placed in a separate paragraph with a separate subtitle. 

564, 573, 667 etc.
Please check Table S1, S2, and S3 references, since they are replaced at several places. 
e.g., at 594: it must be Table S2 instead of Table S1.

Author Response

We thank the reviewer for her/his interesting comments/corrections, 

here it is a point-bu point-response:

-389
The fluence rate (mW/cm2) and some characteristic treatment times (s) should be given to see the progress of the total light dosage (TLD) calculation and to make the calculation process comparable to other jobs. 

REPLY: The most suitable value to allow the comparison with other works is the Total light dosage given in J/cm2 as it is indicated in the equation (page 9 line

-393 in the current version, experimental section).

REPLY:The treatment time depends on the power (fluence rate) of each light irradiation, which is in the range of 12 mW to 25 mW, and the treatment time is varied accordingly to reach a required TLD doses, being in all the cases shorter than 30 min.

-394
The scheme or description of the geometric arrangement of the light irradiation should be given so that they can be compared to the cell irradiation arrangement of others.

REPLY:We have added a picture of the light irradiation set up as Figure S6 in the supplementary information section. The rest of figures in the ESI have been renumbered

-467
The lowest zeta potential here is -37. Therefore, either the absolute value of the zeta potentials should be used in the description, or if an absolute zeta potential is used, the sentence should be reworded accordingly to correct the wording.

REPLY:The reviewer is right and to avoid misinterpretation we have rewrite the sentence page 12 line 488: “According to zeta potential (Table 2), the least favored value (-4.3 mV) was registered for sample RB-PEG750-NP-a with the shortest PEG chain…..”

-487
"The stability of the nanoparticles can also be studied by the absorption spectra of the RB-PEG-MSNs samples in water suspension (Figure 2)."
It should be explained that statement how the stability of the compounds can be deduced from absorption experiments. It requires a more detailed explanation and relevant references (a textbook, a review, or several explanatory articles). Especially it would be necessary since, in the figure, the RB is not in the water, as the statement indicates, but in methanol.

REPLY: We have added in figure 2 (page 12) the absorption spectrum of Rose Bengal in water for a better comparison with the Rose Bengal covalently linked to the silica particles in aqueous suspension. The changes in the absorption band are usually related with dye aggregation, which induce changes in the shape of the absorption. Briefly, different geometries adopted by the dye monomer in the dimer can form new bands or shoulders which can appear blue-shifted or red-shifted respect to the main absorption band. In reference 107 (included in the text) these facts are discussed.

-324
There is no section attributed to the confocal microscopy applied in the materials and methods. Please, in the manuscript, describe the confocal imaging conditions used.

REPLY:The reviewers is completely right. We have added a new section 2.6.3. confocal microscopy dedicated to confocal microscopy description (page 9, lines 399-410)

-541
If confocal images were made, please indicate the depth of the image sections in micrometers. If confocality was not applied, please indicate it appropriately here.

REPLY:Confocality was not applied to obtain the images showed in the manuscript. Images were taken at the same depth in both brightfield and fluorescence channels. We have clarify it in the revised version. (section 2.6.3) 

-544
The results and discussion section would be better if the parts would be emphasized with subtitles and placed in separate paragraphs. In its current form, experiment and statements are very converging. The reader's attention would be better guided if the more important parts were placed in a separate paragraph with a separate subtitle. 

REPLY:Thanks for the recommendation. Indeed, it is quite long manuscript. Now we have divided into sub-sections by adding subheading to make the reading more clear and easy.

3.1 Silica nanoparticles characterization (page 10, line 425)

3.2. Optimization of the functionalization of silica nanoparticles with rose Bengal as PS. (page 11, line 473)

3.3. Photosensitized silica nanoparticles with other photosensitizers (page 14, 577)

3.4. In vitro experiments in HeLa cells (page 16, line 630)

-564, 573, 667 etc.
Please check Table S1, S2, and S3 references, since they are replaced at several places. 
e.g., at 594: it must be Table S2 instead of Table S1.

REPLY: Our apologies for those errors. We have now revised and corrected all those mistakes thoughtout the manuscript

Reviewer 2 Report

So far, many articles on PDT using BODIPY have been reported. The author should cite those articles to clarify the novelty of this manuscript. If the authors synthesize new BODIPY derivatives, the authors should clarify the difference in PDT efficiency from the previously reported BODIPY derivatives.

Author Response

Thanks for your comment. In the present version we have added new sentences and references related with BODIPY in different nanocarrieres and particularly in silica nanoparticles  (lines 72-74 and 98-101)

The new references are now numbered as 26,34-39 y 71-90. Now the readers can have a better knowledge of what is been doing in this research topic. A direct comparison in the PDT efficiency is not possible since the treatment applied (light, exposure time..), cell lines used, etc are different.